# Low-Cost, Real-Time Polymerase Chain Reaction System for Point-of-Care Medical Diagnosis

**DOI:** 10.3390/s22062320

**Published:** 2022-03-17

**Authors:** Tchamie Kadja, Chengkun Liu, Yvonne Sun, Vamsy P. Chodavarapu

**Affiliations:** 1Department of Electrical and Computer Engineering, University of Dayton, 300 College Park, Dayton, OH 45469, USA; kadjat1@udayton.edu (T.K.); liuc30@udayton.edu (C.L.); 2Department of Biology, University of Dayton, 300 College Park, Dayton, OH 45469, USA; ysun02@udayton.edu

**Keywords:** polymerase chain reaction (PCR), fluorescence sensing, low-cost PCR, proportional–integral–derivative (PID) controller, point-of-care diagnostics, COVID-19, food quality, water quality

## Abstract

Global health crises due to the prevailing Coronavirus Disease 2019 (COVID-19) pandemic have placed significant strain on health care facilities such as hospitals and clinics around the world. Further, foodborne and waterborne diseases are not only spreading faster, but also appear to be emerging more rapidly than ever before and are able to circumvent conventional control measures. The Polymerase Chain Reaction (PCR) system is a well-known diagnostic tool for many applications in medical diagnostics, environmental monitoring, and food and water quality assessment. Here, we describe the design, development, and testing of a portable, low-cost, and real-time PCR system that can be used in emergency health crises and resource-poor situations. The described PCR system incorporates real-time reaction monitoring using fluorescence as an alternative to gel electrophoresis for reaction analysis, further decreasing the need of multiple reagents, reducing sample testing cost, and reducing sample analysis time. The bill of materials cost of the described system is approximately $340. The described PCR system utilizes a novel progressive selective proportional–integral–derivative controller that helps in reducing sample analysis time. In addition, the system employs a novel primer-based approach to quantify the initial target amplicon concentration, making it well-suited for food and water quality assessment. The developed PCR system performed DNA amplification at a level and speed comparable to larger and more expensive commercial table-top systems. The fluorescence detection sensitivity was also tested to be at the same level as commercially available multi-mode optical readers, thus making the PCR system an attractive solution for medical point-of-care and food and water quality assessment.

## 1. Introduction

Since 2020, the world has been experiencing a severe Coronavirus Disease 2019 (COVID-19) pandemic caused by the Severe Acute Respiratory Syndrome Coronavirus 2 (SARS-CoV-2). Across the world, local, state/provincial and central governments have struggled to address the extraordinary human, economic, and social tragedy caused by the pandemic. The current pandemic revealed the vulnerabilities in healthcare systems where, in addition to critical shortages of healthcare personnel, ventilators, and personal protective equipment, there is a widespread need for diagnostic tools to quickly assess patient health. Medical equipment such as Polymerase Chain Reaction (PCR) systems have played a crucial role during the COVID-19 pandemic. The PCR test for COVID-19 has been frequently used to analyze the upper respiratory specimens from people, looking for genetic material (ribonucleic acid or RNA) of SARS-CoV-2. The pandemic has caused a dire impact on all countries, but emerging and resource-poor countries which lack adequate healthcare infrastructure have faced more difficulties. Here, low-cost and simple-to-use PCR systems would allow all countries to better handle current and future health crises by quickly testing the patient samples.

In recent years, prevalence in foodborne illnesses caused by pathogens, such as *E. coli* O157:H7, *Salmonella*, *Listeria monocytogenes*, and *Campylobacter*, have affected large populations, causing up to 420,000 deaths yearly, of which 30% are associated with children under five years old [1,2]. Drinking water wells used in many parts of the world are also a major source of bacterial contamination [2,3,4]. Norovirus is a leading cause of epidemic and endemic acute gastroenteritis worldwide, affecting children particularly [5,6]. These situations show the need for low-cost, real-time and rapid bacterial testing in food and drinking water.

Since its introduction in the 1980s, PCR has transformed the fields of biological sciences and medicine [7]. PCR is an enzymatic assay that allows for the amplification of a specific DNA fragment from a complex pool of DNA [8]. It plays a key role in many fields including pathogen analysis, environmental monitoring, and food inspection [9,10,11,12]. Commercial PCR systems are expensive and large in size [13,14,15], with prices from $4000 for simple machines to more than $90,000 for real-time PCR systems [16]. Various types of low-cost PCR systems have been developed previously [17,18,19,20,21,22,23]. Further, many miniaturized system-on-chip and microfluidics technologies have been used to develop PCR systems [24,25]. Most low-cost designs developed to date have limited fixed two-step or fixed three-step temperature settings used for thermal cycling, while others would require reprogramming and altering the system hardware configuration if changes in the protocol were to be made [17,18,20,21]. Other designs lack precise temperature control, which makes them unsuitable for clinically relevant sensitive and selective reactions [17,21]. Moreover, many low-cost PCR designs do not incorporate real-time reaction monitoring with built-in fluorescence detection capabilities, which would require the use of time-consuming gel electrophoresis. Real-world assays require flexible user-defined parameters, such as number of steps, hold time, step temperature, and number of cycles.

We describe the development of a practical, cost-efficient, simple-to-use, portable, and real-time PCR system. The proposed design uses a metallic thermal block and heating lid salvaged from an old PCR system, which if needed could be easily remade with 3D printing. Other parts of the system included commercial off-the-shelf items. The complete system is assembled in a custom-made, low-cost, 3D-printed enclosure. The current system version and sample protocol has been tested with DNA samples. Verification of the system functionality with RNA samples, which would be needed for COVID-19 surveillance, is underway. *Listeria monocytogenes*, a Gram-positive bacterial pathogen, was used as a test sample in the current work to assess the capability of our system, given its threats to food safety [26]. Being able to quickly detect the bacteria at a low number of amplicons will be useful to examine the functionality of our developed system. The test samples are analyzed using the developed PCR system, without any sample pretreatment. The manuscript is organized as follows: In Section 2, we present an overview of the materials and methods, followed by results and discussion in Section 3. Section 4 describes the conclusions.

## 2. Materials and Methods

### 2.1. System Architecture

The PCR system presented in this work has three main parts: (i) a thermal cycler, (ii) an optical read-out system, and (iii) a network device.

The temperature cycler was built using a high-throughput PCR architecture. The parts consisted of a 24-well metallic thermal block incorporating two heaters of 70.3 Ω and 68.7 Ω, respectively. The system has a 14.7 Ω (20 °C)–20.93 Ω (105 °C) heated lid. Both parts were obtained from an old and discarded PCR device GeneAmp 2400 (Applied Biosystems, Foster City, CA, USA). The thermal block holds PCR reaction tubes while serving as the primary heating component. The block is heated using a 67 V, 2 A DC power supply. Meanwhile, the cooling process during thermal cycling is achieved using two thermoelectric cooler (TEC) Peltier modules CP105433H (CUI Devices, Lake Oswego, OR, USA) of size 40 mm × 40 mm × 3.3 mm, in conjugation with a heat sink 394-2AB (Wakefield-Vette, Nashua, NH, USA) of size 139.7 mm × 38.1 mm × 127 mm. Here, the TECs would work in junction with the thermal block to heat the specimen during the high temperature phase. TECs are powered with a 12 V, 20 A DC power supply, which also serves as power source for the rest of the system. High-performance thermal paste (Corsair XTM50) was used for maximum heat conduction between all thermal cycler objects. Two fans (120 mm × 120 mm × 25 mm, 12 V) were added to the system to keep the heat sink from overheating and to accelerate the cooling process. A structural diagram of the path of heat flow between various objects is shown in Figure 1.

Real-time PCR, also known as Quantitative PCR (qPCR) is achieved by the detection of emitted light from an excited fluorophore which is embedded in the DNA amplicons [27] during the PCR annealing phase, using an LED as excitation source. This allows for real-time DNA assay without relying solely on gel electrophoresis. The proposed low-cost optical system is composed of an LED as light source, with a 3D-printed receptacle that can hold a PCR tube, an optical filter, and a chip-scale spectrometer. In operation, the incoming excitation light travels from underneath the PCR tube, while the emitted light is detected at 90° from the excitation spectrum to minimize background noise. The emission wavelength is detected using a low-cost chip-scale 10-channel spectrometer (AMS AG, model: AS7341) with an orange optical filter (546 nm center wavelength, Full Width at Half Maximum (FWHM) of 8 nm) in the foreground. The spectrometer utilizes a light sensor that can detect center wavelengths at 415 nm, 445 nm, 480 nm, 515 nm, 555 nm, 590 nm, 630 nm, and 680 nm (all with FWHM of 8 nm), which is useful for different types of fluorescence applications. Figure 2 shows the diagram of the optical system.

Real-time fluorescence detection data from the photodetector was transmitted to the user via the network module. A low-cost ESP8266 WiFi module was used in the system design. This setting makes it compatible with most of the current WiFi compatible devices such as phones, computers, and tablets.

In the proposed system, the temperature tracking, heating, cooling, and data transmission is performed under the control of a low-cost 8-bit RISC-based ATmega2560 microcontroller. The microcontroller incorporates a multichannel 10-bit Analog to Digital Converter (ADC) as well as an 8-bit Pulse Width Modulation (PWM) capability. This microcontroller was used as part of an Arduino Mega 2560 board for programming purposes and to take advantage of the 16 MHz embedded clock oscillator offered by the board. In that regard, 490 Hz PWM signals can be generated and used for precise temperature control of the thermal block and lid. At such high frequency, and because the pulse rate is higher than the rate of temperature variation, only the average PWM voltage becomes relevant. In addition, power MOSFETs were used to control the electrical current flow through each conductor.

Temperatures of the thermal block and lid are measured by acquiring the resistance of the embedded thermistors in both parts. For a heater with known conductor material, the resistance as a function of temperature is given as,
(1)$$R\left(T\right)=R\left(T_{0}\right)\left[1+\alpha \Delta T\right],$$
where,

R(T)= Conductor resistance at temperature T,

$R\left(T_{0}\right)=$ Conductor resistance at reference temperature $T_{0}$

α= Temperature coefficient of resistance for conductor material

T= Conductor temperature in degrees Celsius

$T_{0}=$ Reference temperature at which α is specified

ΔT= Temperature difference between Conductor temperature and reference temperature

Temperature can be derived from the previous expression for a measured resistance value.

A custom Printed Circuit Board (PCB) holds the main electronic components, and the whole system fits in a 3D-printed enclosure. The overall dimensions are 21 cm × 16 cm × 20 cm and the unit weighs 500 g.

Figure 3 shows the exterior design of the system, and Figure 4 illustrates the schematic diagram of the full system architecture.

### 2.2. System Configuration

#### 2.2.1. Control System

First, a resistance–temperature curve was generated for the metallic thermal block that was harvested from a GeneAmp PCR System 2400 and the data was used to build a look-up table. As mentioned previously, the thermal block could either be harvested from an old PCR system or custom manufactured. The resistance was recorded using a benchtop ohmmeter and its corresponding temperature value was acquired with an infrared thermometer. The curve of the temperature as a function of resistance has a logarithmic trend with a coefficient of correlation of 0.98. The resistance–temperature characteristics of the thermal block are shown in Figure 5.

The equation of the resistance–temperature characteristics curve was obtained as,
(2)T=−25.31ln(R)+314.68, 

PCR thermal cycling requires precise temperature control to achieve optimal results. In that regard, a novel Proportional–Integral–Derivative (PID) scheme was used as the thermal control system. Because the system could be characterized as not “well-behaved” due to the extended delay in reaction time, regular tuning techniques such as Ziegler–Nichols and Cohen–Coon are not suitable [28,29]. Using traditional PID control for a PCR reaction would imply switching between temperature setpoints and making the system react accordingly. Using such controllers on a highly delayed system would fail to meet the typically 30 s (or less) PCR step requirement because of the elongated settling time. For that purpose, we propose a new method called the Progressive Selective PID Controller (PSPC), to overcome the time delay limitation.

In our method, three PID controllers are executed based on a selector command. Each PID controls the denaturation, the annealing, and the extension phase, respectively. The control variable u(t) of each controller is saved in memory and progressively updated as the process continues. This prevents loop recalculation of PID values and allows continuous fine tuning. Therefore, the system becomes more precise over time. The initial controller setting was performed using a heuristic approach, depending on the type of material used. The system can be characterized as a second-degree PID control system. In regular control, the PID controller can be modeled as,
(3)$$u\left(t\right)=K_{p}e\left(t\right)+K_{I}\int_{0}^{t}e\left(t\right)dt+K_{D}\frac{de\left(t\right)}{dt}.$$

The transfer controller transfer function can be approximated as [30,31],
(4)$$C\left(s\right)=\left[K_{p}\;K_{I}\;K_{D}\right]\left[\begin{array}{c} 1\\ \frac{1}{S}\\ \frac{S}{\tau S+1} \end{array}\right]$$

By defining,
$$\theta^{T}=\left[K_{p}\;K_{I}\;K_{D}\right],$$
(5)$$\emptyset_{1}=\left[\begin{array}{c} 1\\ \frac{1}{S}\\ \frac{S}{\tau S+1} \end{array}\right]$$
we can rewrite (2) as,
(6)$$C\left(s\right)=\theta^{T}\emptyset_{1}$$

With our improved PID controller, the controller transfer function can be redefined as,
(7)$$C\left(s\right)=\left(SLR==D\right)\left(\theta_{D}^{T}\emptyset_{1}\right)+\left(SLR==A\right)\left(\theta_{A}^{T}\emptyset_{1}\right)+\left(SLR==E\right)\left(\theta_{E}^{T}\emptyset_{1}\right)$$

With,

SLR= selector value, D= Denaturation step, A= Annealing step, E= Extension step,

$\theta_{n}^{T}=$ PID gains at specified selector value

Figure 6 shows the temperature controller schematic diagram.

#### 2.2.2. Photodetection Procedure

Fluorescence measurements can be performed during thermal cycling after pausing the process at one of its stages, removing the control PCR tube from the heating block, and placing it into the optical system. Performing such measurement during the extension stage will be optimal, as the temperature is low enough to prevent evaporation inside the tube after removing the heated lid, as well as suitable to further allow new DNA strands formations. Prolonged exposure to an excitation source has adverse effects on the fluorescence potency of the fluorophore used for real-time PCR monitoring [21,32]. This effect, known as photobleaching, is intensified by sub-optimal irradiance [33]. The optical system used in our design has a controlled light emission of 3 s and uses optimal light intensity to reduce photobleaching effects. The various fluorophores used in PCR have different excitation wavelengths, which accordingly determine the required excitation LED.

#### 2.2.3. Quantitative DNA Analysis by PCR

Primers with longer amplicon size tend to yield less amplification efficiency because there is need for more time to complete DNA elongation, therefore they reach the reaction plateau at later cycles (high *C_T_*). Using that property, it was determined experimentally that with primers of product size between 300 and 600 base pairs (bp), and a PCR protocol of 3 steps and 40 cycles, starting point amplicons with concentrations at different orders of magnitude yield final DNA concentrations at a gradient. This protocol was used to quantify listeria concentration in water by applying fluorescence on the PCR final products.

### 2.3. System Automation and Versatility

As mentioned previously, PCR reaction settings largely depend on the type of DNA and expected outcome. Using fixed temperatures in thermal cycling would not work well for all assays and must be tailored accordingly. The PSPC was designed to allow user input and custom-made experiments. Users can define the number of cycles, number of steps for each cycle, the step temperatures, step hold times, and the system will automatically adapt without affecting its performance. In addition, that PSPC was designed to operate autonomously with minimal human interaction during the process. The autonomy and versatility of the system were achieved by developing a C++ program for the ATmega2560 microcontroller that would allow processing the thermal cycling and data communication. An on-board user-friendly interface with keypad, buttons and display is provided for input/output interaction without the need for an external device or computer, even though the user could also remotely access real-time fluorescence data on an external WiFi connected device. The remote data is provided and presented via an automatically generated HTML interface on the device connected to the PSPC.

### 2.4. Bill of Materials

The current system was designed to be a cost-effective solution with use in resource-poor countries. The bill-of-materials cost (see Table 1) for the system was about $340.

### 2.5. Gel Electrophoresis Analysis

Following the PCR reactions, amplicon quantification was performed using a 1% or 1.2% gel (100 mL Tris-Borate-EDTA (TBE) buffer, 1 g or 1.2 g agarose) locally prepared for the experiment, run at 100 volts for 30 min, loading 2 µL 6X loading dye and 10 µL PCR sample. The ladder used was a 1 kb DNA ladder (New England Biolabs, cat. no. N3232L). The images were captured using UVP Multispectral Imaging System (Biospectrum 500, LM-26, BioChemi 500 Camera f/1.2).

### 2.6. PCR Conditions

The experiments described in this work compare the efficiency of our PSPC system versus a commercially available PCR system (Applied Biosystems, model: StepOnePlus). The developed PCR thermal cycling was conducted concurrently on PSPC and the StepOnePlus systems. Fluorescence measurement was compared with the commercial device Synergy LX multi-mode reader (BioTek Instruments, model: SLFXA) with green filter cube loaded (BioTek 1505005, Ex 485/20, M 510, EM 528/20).

Stock primers had a 100 µM concentration and were diluted to a final working stock of 20 µM. All the reaction volumes are 20 µL on both the PSPC and StepOnePlus. We used 0.2 mL tubes in the PSPC as they allow better adhesion with the heated lid. Tubes used on both PSPC and StepOnePlus were low-profile polypropylene tubes, although capillary glass tubes could be used as well. Fresh *Listeria monocytogenes* colonies were collected from a streaked plate and placed in sterile water. The initial bacterial solution (Tube 1) was diluted further in a serial 1:10 dilution down to 1:10,000,000 (Tube 8) diluted solution. Tubes 8 and 7 were plated on LB agar plates for colony count, and count results could be extrapolated for higher concentrations. Amplicon concentrations were mixed with Select MasterMix (Applied Biosystems SYBR, Mfr. No. 4472908) and combined with different primer sets (each at 20 µM working concentrations). Table 2 summarizes the target, amplicon size, primer sequence, and reaction reagents for gel electrophoresis analysis and fluorescence analysis, respectively. The PCR protocols were identical on both PSPC and StepOnePlus systems, which were as follows: 10 min hot-start at 95 °C, followed by 40 cycles of 95 °C (30 s denaturation), 55 °C (30 s annealing), and 72 °C (60 s extension), followed by 10 min final elongation at 72 °C.

## 3. Results

### 3.1. Temperature Curve Analysis

The heating and cooling speed recorded for the PSPC was comparable to the StepOnePlus system. With the thermal block fully loaded (24 tubes), we recorded an average of 40 s to cool down from 95 °C to 55 °C, 9 s to heat up from 55 °C to 72 °C, and 13 s from 72 °C to 95 °C. Table 3 summarizes the thermal transition rate between PCR steps at the preset temperatures. For a 40-cycle reaction, it took an average of 2 h 23 min on PSPC, versus 2 h 10 min on the StepOnePlus system. Table 3 summarizes the heating and cooling rates of the system.

With the selective progressive controller, the temperature curve shows low slope and reduced fluctuations at each temperature step, which is appropriate for temperature selective applications. Figure 7 shows the temperature curve recorded for the PSPC at 95 °C–55 °C–72 °C cycle. Figure 8 shows the setpoint error at different cycles for the extension step.

The error at different cycles was compared using the area under the curve at each of the cycles. The definite integral computation resulted in values of 14,675.24, 8552.74, and 545.32 for cycle 1, cycle 2, and cycle 3, respectively. This shows how the system improves at tracking the reference point over time. This is the result of the– progressive selective control.

### 3.2. Colony PCR Amplification for Listeria Detection

We used PSPC for listeria detection by amplifying DNA from a listeria colony as described in the PCR conditions. The total runtime was 2 h and 23 min for PSPC for 40 cycles (30 s, 30 s, 60 s),10 min hot-start, and 10 min final elongation. Because the heating block is not fully enclosed, a little condensation and evaporation could be observed in the exposed area of tubes, but it does not adversely affect the reaction outcome. Figure 9 shows the gel electrophoresis capture with both PSPC and commercial reaction amplicons for tubes 5 to 8 (1/10,000 to 1/10,000,000 dilution). Colony count from plated bacteria gave an average bacterial presence of 280 CFU/mL in tube 8, and 2160 CFU/mL in tube 7. This result can be extrapolated at higher concentrations to 20,000 CFU/mL in tube 6 and 200,000 CFU/mL in tube 5. The PCR result shows that the right amplicon sizes were amplified, and results are comparable on both systems. Even in instances such as 629 bp product size, the PSPC produced a brighter gel signal compared to the StepOnePlus system.

Longer product size amplicon amplification was not successful on both systems as the process would require a longer elongation time than the one used in the protocol. Nevertheless, the other results were similar on both systems.

### 3.3. Bacterial Count Using Fluorescence

PSPC was used to correlate *Listeria* amplicon population, or colony counts, in water, using the aforementioned PCR protocol. A 2 s integration time was used on the sensor for fluorescence measurement. Colony count shows an estimate of less than 10 CFU per PCR tube for 1/10,000,000 dilution. Figure 10 and Figure 11 show the summary of the fluorescence level as a function of amplicon population on the StepOnePlus and PSPC systems.

The parallel operation on the StepOnePlus and PSPC systems showed similar level of sensitivity. Fluorescence signal recordings show a gradient from tubes 1 to tube 4 (1/1 to 1/1000 dilution) in both settings. Below that threshold, the detection no longer follows a gradient pattern.

The experiment did not perform any type of bacteria preservation method, which could explain a random decay of amplicon population at lower concentrations. Also, the negative control showed a higher fluorescence level as compared to samples with lower bacteria concentrations. These suggest that the reagents used in the reactions contain background fluorescence.

## 4. Conclusions

We described the design, development and testing of a portable, low-cost, and real-time PCR system that can be utilized for food and water quality assessment. In addition, the system can be used in emergency health crises and point-of-care diagnostics. The described PCR system incorporated real-time reaction monitoring using fluorescence detection and eliminated the need of gel electrophoresis for reaction analysis, further decreasing the need of multiple reagents, reducing sample testing cost, and reducing sample analysis time. The bill of materials cost of the described system was approximately $340. The described PCR system utilized a novel progressive selective proportional–integral–derivative controller that helps in improving system accuracy and reducing sample analysis time. In addition, the system employs a novel primer-based approach to quantify the initial amplicon target concentration. The system is versatile and provided a clinically relevant performance, and can be used with a wide range of DNA samples. The present work focused on DNA, but the instrument could easily be adapted to RNA-type surveillance by incorporating an additional cDNA synthesis step prior to PCR.

## Figures and Tables

**Figure 1 sensors-22-02320-f001:**
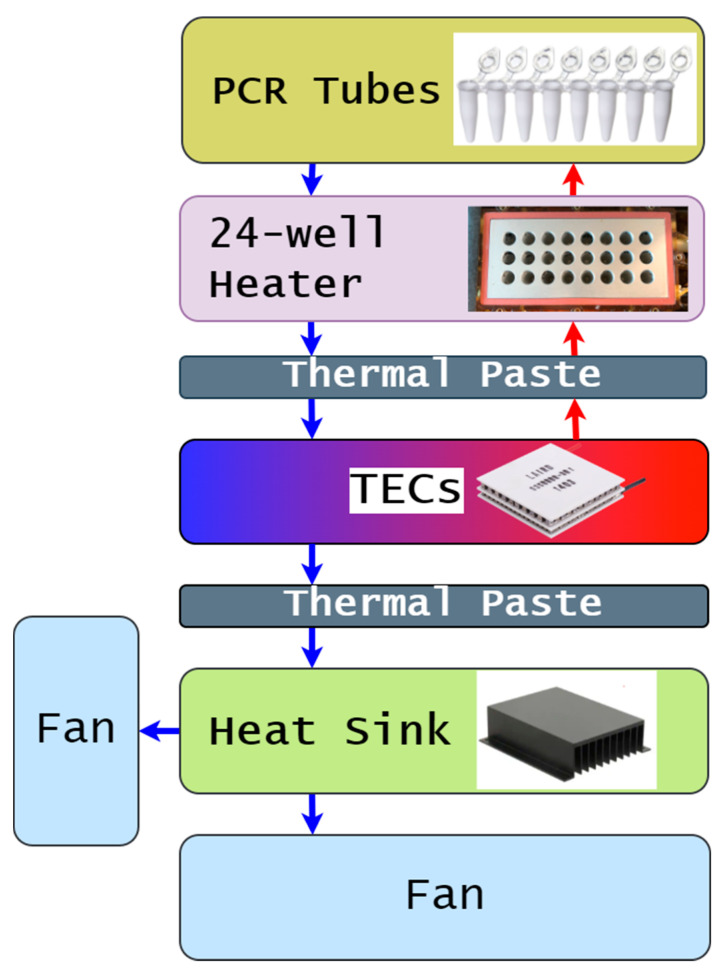
Thermal flowchart of heat transfer between temperature cycling objects. In blue: cooling stage. In red: heating stage.

**Figure 2 sensors-22-02320-f002:**
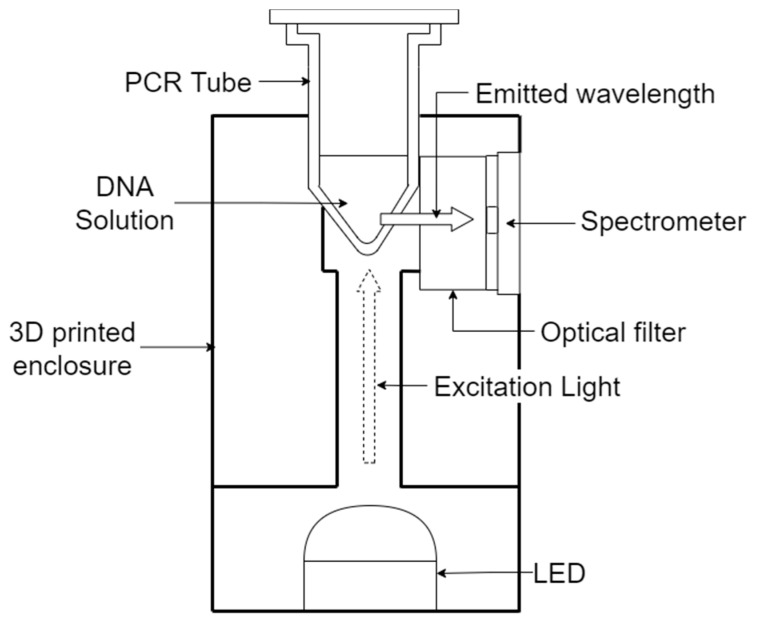
Schematic of the optical system.

**Figure 3 sensors-22-02320-f003:**
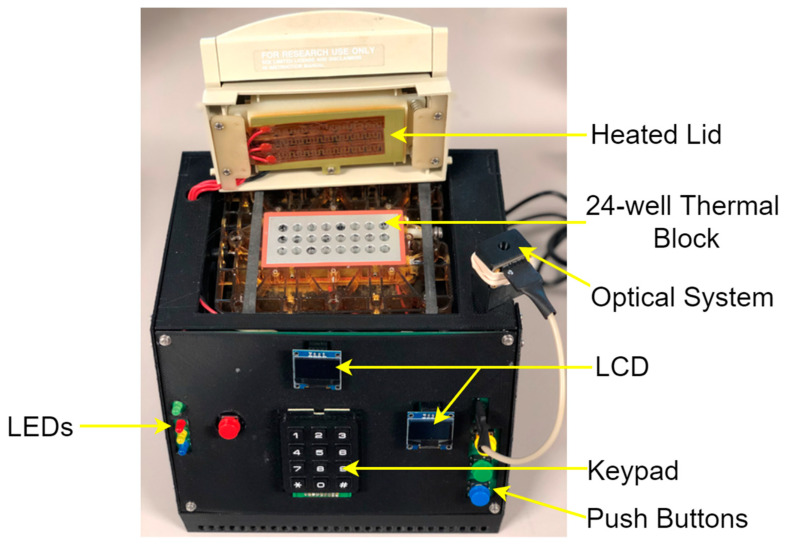
Exterior appearance of the system.

**Figure 4 sensors-22-02320-f004:**
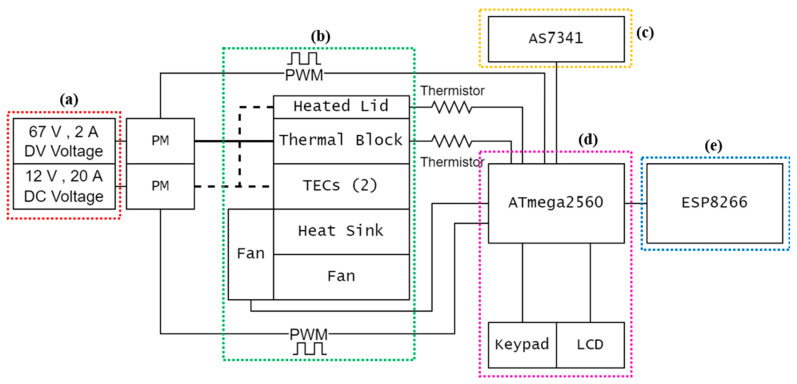
Structural diagram of PCR system. (**a**) Power supply; (**b**) Thermal control; (**c**) Optical system; (**d**) Control Unit; (**e**) Communication module.

**Figure 5 sensors-22-02320-f005:**
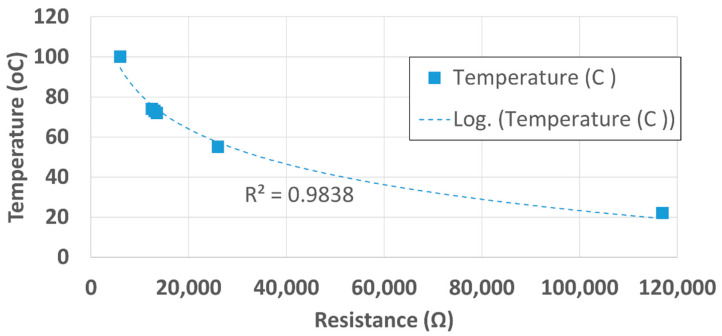
Resistance–temperature characteristics curve of thermal block.

**Figure 6 sensors-22-02320-f006:**
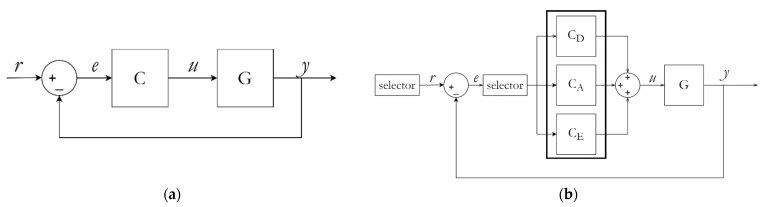
(**a**) Conventional PID controller. (**b**) Proposed PSPC temperature controller.

**Figure 7 sensors-22-02320-f007:**
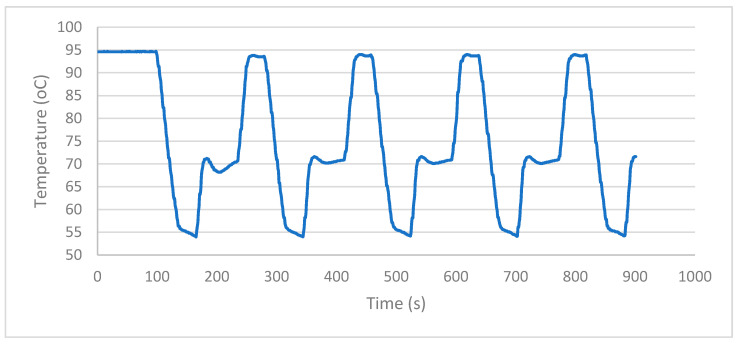
Temperature curve of PSPC.

**Figure 8 sensors-22-02320-f008:**
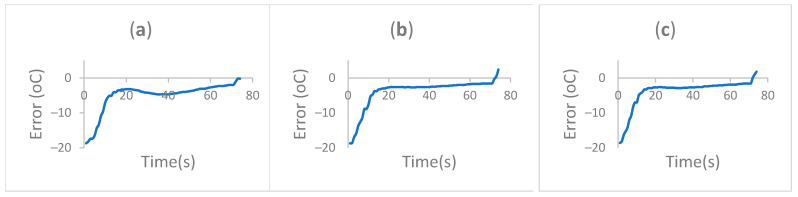
Extension (72 °C) setpoint error comparison over different cycles. (**a**) Cycle 1. (**b**) Cycle 2. (**c**) Cycle 3.

**Figure 9 sensors-22-02320-f009:**
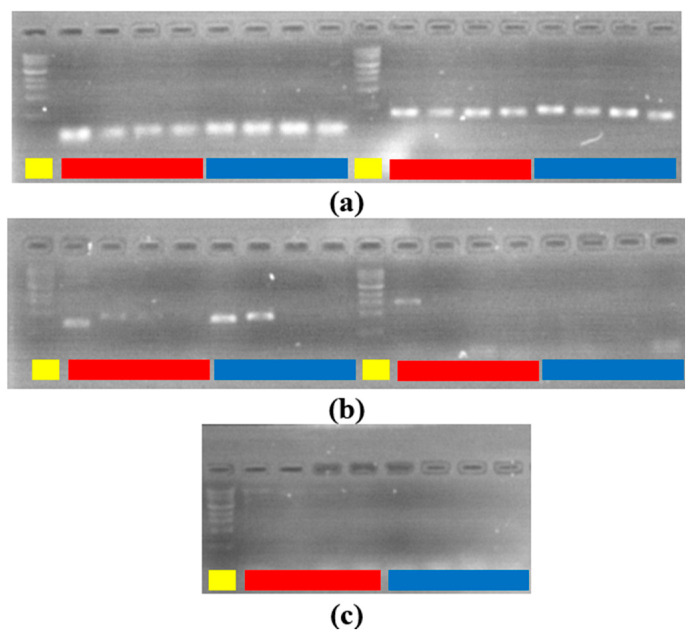
Gel electrophoresis results. yellow: ladder. red: commercial. blue: PSPC. (**a**) left: 115 bp tube 5 to 8. right: 320 bp tube 5 to 8. (**b**) left: 629 bp tube 5 to 8. right: 1054 bp tube 5 to 8. (**c**) 1456 bp tube 5 to 8.

**Figure 10 sensors-22-02320-f010:**
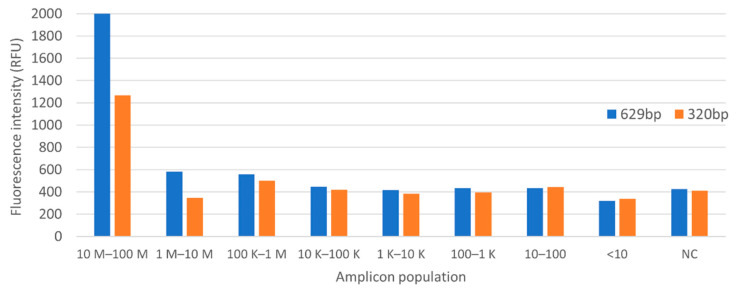
Fluorescence intensity on StepOnePlus system.

**Figure 11 sensors-22-02320-f011:**
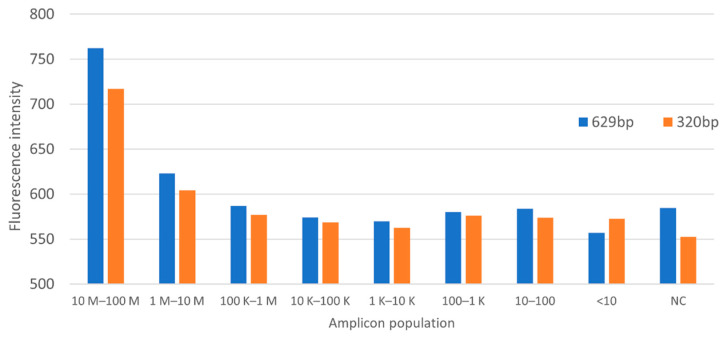
Fluorescence intensity on PSPC.

**Table 1 sensors-22-02320-t001:** Main parts used in PSPC design.

Parts	Quantity	Cost (Total)
Peltier Plates	2	$63.14
Heat Sink	1	$27.96
ESP 8266 Microcontroller	1	$6.49
ATMega 2560 Microcontroller	1	$15.99
AS7341 Spectrometer	1	$15.95
Custom PCB	1	$12.62
LCD	2	$13.98
67.2 V, 2 A Power Adapter	1	$18.99
30 A Relay	1	$12
12 V, 20 A Power Adapter	1	$41.99
Fan	2	$25.98
Keypad	1	$4.50
1 Kg PLA filament	1	$17.99
LED (Light Source)	1	$13.99
Optical Filter	1	$38.60
**Heated Lid**	1	-
**Thermal Block**	1	-
Miscellaneous		$10
**Total cost**		$340.17
**Manufacture/Assembly time**		24 h (mostly 3D printing)

**Table 2 sensors-22-02320-t002:** PCR target, analysis type and reaction components.

Amplicon Size	Analysis Type	Reagents
Listeria (115 bp)	Gel electrophoresis	10 µL Select Master Mix
		1 µL F. primer 5′- CAA GCG TTG TCC GGA TTT ATT G -3′
		1 µL R. primer 5′- GCA CTC CAG TCT TCC AGT TT -3′
		1 µL template
		7 µL deionized H_2_O
Listeria (320 bp)	Gel electrophoresis	10 µL Select Master Mix
		1 µL F. primer 5′- GGT GGA GCA TGT GGT TTA ATT C -3′
		1 µL R. primer 5′- TTC GCG ACC CTT TGT ACT ATC -3′
		1 µL template
		7 µL deionized H_2_O
Listeria (629 bp)	Gel electrophoresis	10 µL Select Master Mix
		1 µL F. primer 5′- GTA GCG GTG AAA TGC GTA GA -3′
		1 µL R. primer 5′- GCC TAC AAT CCG AAC TGA GAA TA -3′
		1 µL template
		7 µL deionized H_2_O
Listeria (1057 bp)	Gel electrophoresis	10 µL Select Master Mix
		1 µL F. primer 5′- TGG TTT CGG CTA TCG CTT AC -3′
		1 µL R. primer 5′- CTT CGC GAC CCT TTG TAC TAT C -3′
		1 µL template
		7 µL deionized H_2_O
Listeria (1456 bp)	Gel electrophoresis	10 µL Select Master Mix
		1 µL F. primer 5′- CGA ACG AAC GGA GGA AGA G -3′
		1 µL R. primer 5′- GGC TAC CTT GTT ACG ACT TCA -3′
		1 µL template
		7 µL deionized H_2_O
Listeria (320 bp)	Fluorescence	10 µL Select Master Mix
		1 µL F. primer 5′- GGT GGA GCA TGT GGT TTA ATT C -3′
		1 µL R. primer 5′- TTC GCG ACC CTT TGT ACT ATC -3′
		8 µL template (8 µL deionized H_2_O for negative control)
Listeria (629 bp)	Fluorescence	10 µL Select Master Mix
		1 µL F. primer 5′- GTA GCG GTG AAA TGC GTA GA -3′
		1 µL R. primer 5′- GCC TAC AAT CCG AAC TGA GAA TA -3′
		8 µL template (8 µL deionized H_2_O for negative control)

**Table 3 sensors-22-02320-t003:** Temperature rates of change during cycles.

Stage	Transition Rate
95 °C–55 °C	−1 °C/s
55 °C–72 °C	1.89 °C/s
72 °C–95 °C	1.77 °C/s

## Data Availability

Not applicable.

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
