# Peer review of "Low-Cost, Real-Time Polymerase Chain Reaction System for Point-of-Care Medical Diagnosis"

_sensors, 2022, doi:10.3390/s22062320_

Round 1

Reviewer 1 Report

  1. The abstract and Introduction lead the reader to believe the PCR test is for COVID-19, when it is not. This is a little misleading and the authors should indicate that PCR can be used to test for many pathogens, including coronavirus, but was developed in this example for listeria. Considering the position that this is proposed for resource-poor countries, there is an ethical consideration if people adopt this work without understanding that difference.
  2. Please put a space between the number and the unit for all values, i.e. 67 V, 2 A, consistent with SI format. 
  3. Page 3 Lines 102 and 105 seem to say the TEC is used for both heating and cooling. Since they are unidirectional devices, are two TECs used for heating and cooling? If so Fig. 1 and 4 should show this. 
  4. Provide citation on PCR details, Page 3 Line 113
  5. Page 3, Line 137: pulsed voltage is not sensed as continuous. Please clarify the language - the pulse rate is faster than the rate of temperature change, so thermally only the average value of the PWM matters. 
  6. Fig. 4 - what is the dashed line leading to TEC and Heated Lid? A wire?
  7. Page 8 Line 226 - missing word "one its"
  8. Table 1 - should also include parts from an old PCR machine (heated lid and thermal plate)
  9. Fig. 7 and the sentence after - it is not clear how the "system improves over time". Perhaps a second figure showing the setpoint error over time or a temperature error histogram would show this more clearly.

Reviewer 2 Report

The paper under consideration has demonstrated the PCR system for point-of-care medical diagnostic purposes. The paper is quite interesting and beneficial in the time of pandemic situation. The device is cost-effective and offers a simple and efficient way to analyze the sample. I am in the favor of its publication. I have a few minor points which should be considered before the final acceptance of the paper. 

  • Table 1, also mention the time required to manufacture/assemble this device? And is it device reproducible? What is the detection error from one device to another?
  • I am not sure if the PCR test has been performed on patients. How many people participated in the analysis?
  • Can this device be used to analyze 24 different DNA samples at the same time?
  • In Figures 5 and 7, the unit of degree centigrade (C) should be corrected.

Reviewer 3 Report

Title: Low-Cost, Real-Time Polymerase Chain Reaction System for Point-of-Care Medical Diagnosis

Authors: Tchamie Kadja, Chengkun Liu, Yvonne Sun and Vamsy P. Chodavarapu

Recommendation: Publish after minor revision.

This manuscript by Kadja et al. provides a significant level of development and optimization of a compact PCR system for quantitively analysis of DNA. I am sure that the 1st author has gained a high level of competence in engineering optimization, spectral investigation and biological sensing that is worthy of a PhD thesis. In this regard this work is a smashing success. Due to the high quality of the investigation and the care with which this manuscript was prepared there is little ground for review.

(1). I did find that the references in the Intro section is inadequate and can be improved by including the recent reports on COVID-19 findings.

- On Page 2, Lines 50-51, the first report of COVID should be included: Science 2020, 370, 303−304 (DOI: 10.1126/science.abf0521)

- On Page 2, Lines 56-57, the authors mentioned foodborne diseases, nonetheless, the most well-known one called “norovirus” is missing: (doi:10.1042/BJ20140959)

- On Page 2, Lines 53-55, and 61-62, two recent reference aims to make a real-time and rapid COVID detection using a portable mask should be included: Angew. Chem. Int. Ed. 2022, e202112995 (doi.org/10.1002/anie.202112995); Nat. Biotechnol. 2021, 39, 1366 (doi.org/10.1038/s41587-021-00950-3). These topics are highly related to the aims of this manuscript (i.e., portable and rapid detection) and could generate broader readership and awareness of this contribution.

- On Page 2, Line 74, I would be very careful of speaking the temperature control as one of the major hiderance for PCR development, because room temperature PCR (rt-PCR) has been popularly used and well optimized in many reports: Nat. Commun. 2020, 11, 4812 (doi.org/10.1038/s41467-020-18611-5). However, the advantages of cost-effective and real-time signal detection still hold valid in this contribution.

(2). On Page 3, Line 119, the author should specify the right angle as 90 °.

(3). In Figure 2, my major concern is the feasibility of compact fluorometer dealing with a massive of amplified DNA samples. How would the authors improve the capacity of their detection system for measuring a million of samples in one time? The sampling capacity is always a problem for the gold standard PCR till now.

(4). On Page 8, Line 226, should be “at one of its stages”.

(5). Figure 8, caption, should be “red: commercial”.

(6). In Figure 8, SARS-CoV-2 virus uses RNA as the genetic replication material, however, the authors use DNA as the proof-of-concept example. What difference the authors might observe if they use RNA as the amplification target?

(7). Sadly no real COVID-patient samples either from nasal swap or breath were tested. Although this would massively increase the impact of the work, likely the use of real samples at different stages of diseases is beyond the scope of this work?
